# Glucocorticoid-Induced Hyperinsulinism in a Preterm Neonate with Inherited *ABCC8* Variant

**DOI:** 10.3390/metabo12090847

**Published:** 2022-09-08

**Authors:** Emmanuelle Motte-Signoret, Cécile Saint-Martin, Christine Bellané-Chantelot, Bernard Portha, Pascal Boileau

**Affiliations:** 1Department of Neonatal Intensive Care Unit, Poissy St Germain Hospital, 10 rue du Champ Gaillard, 78300 Poissy, France; 2Biologie de la Reproduction, Environnement, Epigénétique et Développement (BREED), Institut National de Recherche pour l’Agriculture, l’Alimentation et l’Environnement (INRAE), Versailles St Quentin University—Paris-Saclay University, 78350 Jouy-en-Josas, France; 3Department of Medical Genetics, Pitié-Salpêtrière Hospital, Sorbonne University, AP-HP, 75013 Paris, France; 4Unité de Biologie Fonctionnelle et Adaptive, Laboratoire B2PE, CNRS UMR 8251, Campus Grands Moulins, Université Paris Cité, 75205 Paris, France; 5Physiologie et Physiopathologie Endocriniennes, Inserm, Paris-Saclay University, 94276 Le Kremlin-Bicêtre, France

**Keywords:** neonatology, diabetes mellitus, K^+^ channels

## Abstract

Glucose homeostasis is a real challenge for extremely preterm infants (EPIs) who have both limited substrate availability and immature glucose metabolism regulation. In the first days of life, EPIs frequently develop transient glucose intolerance, which has a complex pathophysiology that associates unregulated gluconeogenesis, immature insulin secretion, and peripheral insulin resistance. In this population, glucocorticoid therapy is frequently administrated to prevent severe bronchopulmonary dysplasia. During this treatment, glucose intolerance classically increases and may lead to hyperglycemia. We report a case of neonatal hypoglycemia that was concomitant to a glucocorticoids administration, and that led to a congenital hyperinsulinism diagnosis in an EPI with a heterozygous *ABCC8* variant. The variant was inherited from his mother, who had developed monogenic onset diabetes of the youth (MODY) at the age of 23. *ABCC8* encodes a beta-cell potassium channel unit and causes congenital hyperinsulinism or MODY depending on the mutation location. Moreover, some mutations have been observed in the same patient to cause both hyperinsulinism in infancy and MODY in adulthood. In our case, the baby showed repeated and severe hypoglycemias, which were undoubtedly time-associated with the betamethasone intravenous administration. This hyperinsulinism was transient, and the infant has not yet developed diabetes at three years of age. We take the opportunity presented by this unusual clinical presentation to provide a review of the literature, suggesting new insights regarding the pathophysiology of the beta-pancreatic cells’ insulin secretion: glucocorticoids may potentiate basal insulin secretion in patients with *ABCC8* mutation.

## 1. Introduction

Extremely preterm infants (EPIs; defined by World Health Organization as born before 28 weeks postmenstruation) frequently experience early and transient hypoglycemia in the first two days of life [1] primarily due to both limited substrate availability and immature glucose metabolism regulation. Glucose homeostasis at birth is challenging in newborns from mothers with gestational diabetes because the maternal hyperglycemia causes chronic hyperinsulinism [2,3]. After this transition phase, EPIs frequently develop transient glucose intolerance with a complex pathophysiology that associates unregulated neoglucogenesis [4], immature insulin secretion [5], and peripheral insulin resistance [6,7].

These EPIs are at higher risk of bronchopulmonary dysplasia (BPD) and may receive postnatal glucocorticoid therapy in order to withdraw invasive mechanical ventilation. During this glucocorticoid treatment, glucose intolerance classically increases due to induced gluconeogenesis and insulin resistance [8,9], which ultimately leads to hyperglycemia. In addition, bedside care in neonatal intensive care units (NICU) includes the need to follow nutritional guidelines regarding carbohydrates’ and lipids’ enteral and parenteral intakes, in addition to the requirement for closely monitoring capillary glycaemia [10].

Here, we report an original case of neonatal hypoglycemia, which occurred concomitantly with glucocorticoid administration in an EPI with an *ABCC8* mutation that was inherited from his mother, who had gestational diabetes. The hypoglycemia led to a congenital hyperinsulinism diagnosis.

## 2. Case Report

We present the case of a preterm boy born at 27 weeks of gestational age by cesarean section for maternal pre-eclampsia after antenatal administration of glucocorticoids (betamethasone 12 mg IM/24 h twice) to accelerate fetal pulmonary maturation and improve the neonatal outcome of the EPI [11]. The mother was 37 years old, primiparous, and primigravid, and the obstetrical medical record reported a gestational diabetes diagnosis, which required an insulin pomp regimen and a recent HbA1c of 8%.

Birth weight, length, and head circumference were 1065 g (72th percentile), 35 cm (40th percentile), and 26 cm (70th percentile), respectively. The Apgar score was 0–3–8 at 1, 5, and 10 min, respectively. The newborn quickly showed a respiratory distress syndrome, which led to an intratracheal exogenous surfactant administration (poractant, 200 mg/kg) in the first hour of life. Standardized nutritional intakes were provided according established protocols, both by enteral nutrition and early parenteral nutrition through an umbilical vein catheter during 3 days, and then through an epicutaneous cava catheter until day 29.

He presented mild capillary hypoglycemia (mean 30 mg/dL, corresponding to 1.7 mmol/L) during his first 24 h, with an increased glucose intake of 9 g/kg/day in order to maintain glycaemia up to 45 mg/dL. Thereafter, his glycaemia was in the normal range between day 2 and 22 without adding insulin or increasing the recommended intake (glucose and lipids intakes were about 13 and 6 g/kg/d, respectively) (Figure 1). During this period, he did not need any hemodynamic support and presented no early or late-onset neonatal infection.

The neonate developed a severe respiratory disease and still required an invasive mechanical ventilation with a FiO2 up to 0.5 at the end of his third week of life. Thus, we decided to administrate betamethasone intravenously at 0.3 mg/kg/days for 3 days, 0.2 mg/kg/days for 3 days, and 0.1 mg/kg/days for 3 days, according to our NICU protocol. This treatment allowed a withdrawal of the endotracheal tube and definitive non-invasive ventilation with FiO2 0.25 at day 25. The monitoring of capillary glycaemia showed repeated and severe hypoglycemia that began less than six hours after the first betamethasone intravenous injection. In order to normalize the glycaemia, we needed to increase glucose intakes at 16 g/kg/d and lipids intakes at 8 g/kg/d (which includes both enteral and parenteral intakes) (Figure 1). Glycaemia remained under 80 mg/dL despite high caloric intake during the nine days of glucocorticoid treatment, and then stabilized around 100 mg/dL, which allowed a progressive glucose and lipids decrease to their usual range (13 and 6 g/kg/d, respectively, at this age).

We performed hormonal assays to investigate these hypoglycemias. We found extremely high insulin levels, which are associated with high C peptide and pro-insulin levels, thus revealing a hyperinsulinism. Concomitantly, cortisol, growth hormone (GH), and insulin-like factor 1 (IGF1) levels were low. We performed these hormonal assays during the follow-up. We observed a progressive decrease in insulin, C peptide, and pro-insulin levels and a progressive increase in GH and IGF1 levels. Cortisol levels remained low but inhaled glucocorticoids treatment was still administrated daily to the patient (Table 1).

Considering this unusual clinical presentation, we decided to further investigate the mother’s diabetes diagnosis at 23 years of age. She was lean and presented a ketoacidosis and no specific diabetes antibodies, ruling out type 1 and type 2 diabetes. Genetic tests had been performed for the main monogenic diabetic etiologies (*GCK*, *HNF1A*, *HNF4A*, *HNF1B*, *INS*, *ABCC8*, and *KCNJ11*). The DNA sequencing found a heterozygous pathogenic variant in *ABCC8* (NM 001287174.3: c.928G>A, p.Asp310Asn), the gene-coding SUR1, which is one of the K^+^ channels of the beta-pancreatic cells’ subunits. This variant was reported to cause congenital hyperinsulinism with dominant inheritance (CHI) [12,13,14]. As some rare *ABCC8* variants lead to CHI during the neonatal period and diabetes in adulthood [15,16], we theorized this caused her diabetes, despite the fact that she had no hypoglycemia issues in the perinatal period or during previous glucocorticoids treatments for intercurrent viral infections, such as sinusitis.

We found the same heterozygous mutation in our patient, confirming the CHI molecular etiology. After the initial recovery and normalization of glycaemia, we carefully continued his follow-up until the age of three: this infant did not develop diabetes (his last HbA1c was 5.7%), never experienced a symptomatic hypoglycemia recurrence, and had normal growth. The last hormonal assay was performed when he was one year old, and he showed normal concentrations for insulin, C peptide, pro-insulin, GH, and IGF1 (Table 1).

## 3. Discussion

Here, we report a case of congenital hyperinsulinism, which was revealed by glucocorticoids administration in an EPI born from a diabetic mother whose monogenic onset diabetes of the youth (MODY) diagnosis was unknown by the pediatricians. To our knowledge, this is the first described case when glucocorticoids, rather being responsible for hyperglycemia, where administered for the prevention of severe BPD [8,9,17].

Glucose homeostasis is a real challenge for neonates, who have to quickly adapt after birth in order to maintain sufficient glucose production as the continuous glucose infusion by the umbilical cord stops. It is even more challenging for preterm babies who lack energetic substrates to provide efficient gluconeogenesis, and for babies born of diabetic mothers, because they frequently develop a reactional hyperinsulinism [18]. This was initially the case for our patient, who presented mild hypoglycemia during his first day; furthermore, no other particular presentation could have made us think of anything else other than classical EPI early hypoglycemia.

However, EPIs experience hyperglycemia more often and for longer periods because of an altered regulation system [19], which may be increased by glucocorticoid treatment. Glucocorticoids exert massive effects on glucose homeostasis under normal and pathological situations. Chronic administration leads to fast insulin increases [20,21]. This is an adaptive phenomenon of beta-pancreatic cells to normalize the concomitant slight elevation of glycaemia due to decreased glucose peripheral utilization and increased production by gluconeogenesis [22]. This involves numerous signaling pathways, including K^+^ channels, or the alteration of Ca^++^ fluxes [23,24]. In our case, despite a very low gestational age (27 weeks) and the need for extended parenteral nutrition, the patient never presented any hyperglycemia, while previous studies reported a prevalence of hyperglycemia in around 80% of EPIs [25]. Moreover, the glucocorticoid administration was perfectly concomitant (in term of hours) to new episodes of severe hypoglycemia. Knowing that this neonate had a loss-of-function mutation (*ABCC8*), we hypothesize that he had a moderate hyperinsulinism, which may have been countered in his three first weeks of life by the peripheral resistance to insulin characteristic of EPIs [5,6,7].

Other possible regulatory hormones of glucose metabolism need to be investigated in order to rule out an etiology other than hyperinsulinism. In our case, serum cortisol was extremely low during hypoglycemia. The most plausible explanation would be the existence of a glucocorticoid-induced adrenocorticotropin hormone (ACTH) deficiency. GH and IGF1 levels were also low but we previously showed that there is GH resistance in this population [26], and preterm neonates are known to exhibit low IGF1 levels [27]. The progressive normalization of GH and IGF1 levels through the first year of our patient are reassuring regarding the absence of congenital GH deficiency. However, this absence of counter-regulatory hormones for glucose metabolism could only partially explain the recurrence of hypoglycemia, as neonates with CHI have been shown to generate the poorest serum cortisol responses to hyperinsulinemic hypoglycemia [28].

Several recent articles have suggested that glucocorticoids may stimulate beta-cell neogenesis, not only in response to an increased need for insulin due to excessive glucose production, but through a direct effect on calcium influx [29,30,31]. Our ultimate hypothesis would be that this may be increased for beta-pancreatic cells that are very sensitive to electrical activity through K_ATP_ channel mutations, such as in our patient. *ABCC8* and *KCNJ11* encode the subunits SUR1 and KIR6.2 of the beta-pancreatic cell plasma membrane ATP-sensitive potassium (K_ATP_) channel, respectively. This channel has a key role in glucose-induced insulin secretion, linking the metabolic state of the cell to its electrical activity [32]. High plasma glucose concentration levels lead to high ATP/ADP ratios and K_ATP_ channel closure, which triggers insulin release for storage granule via calcium influx [33]. Loss-of-function mutations in SUR1 or KIR6.2 lead to CHI, whereas gain-of-function mutations lead to monogenic diabetes; however, mutation of the same residue can result in either CHI or MODY [16], and some cases have even been described with the same mutation thought to be responsible for CHI in the neonatal period, which subsequently evolves to MODY at an adult age [34]. We did not find any history of hypoglycemia in the patient’s mother. It seems that the only reason our patient exhibited severe hypoglycemia was the betamethasone treatment. We closely followed his evolution, knowing that he was at risk of later developing the same MODY as his mother. We were also attentive to the possibility of hypoglycemia recurrence, especially during infancy illness or in case of glucocorticoid treatment. At the age of three years old, his growth and neurodevelopment were perfectly satisfying, and his HbA1c was normal (5.7%).

We believe that this case is interesting for two reasons. First, it highlights the possibility for this mutation (c.928G>A, p.Asp310Asn) in *ABCC8* to lead to both diabetes and hyperinsulinism, as suggested for other *ABCC8* mutations [16,34]. Second, it suggests a new pathophysiological hypothesis regarding the relationship between glucocorticoids and beta-pancreatic cells, in particular in EPI, a high-risk population for developing metabolic syndrome [35,36]. Moreover, it reminds clinicians to always dig for data: in this case, the cause of the mother’s diabetes’ was missing in her medical record.

## Figures and Tables

**Figure 1 metabolites-12-00847-f001:**
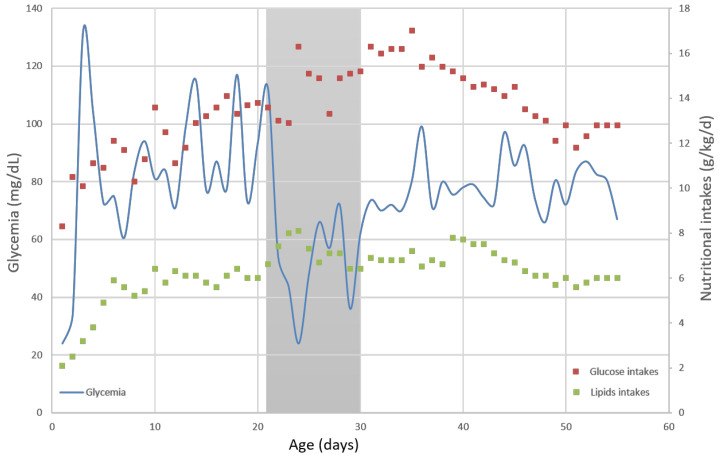
Evolution of capillary glycaemia (mg/dL, on left scale and on blue line) and concomitant nutritional intakes (g/kg/day, on right scale and with red dots for glucose intakes and green dots for lipids intakes) within days. Shaded area represents betamethasone administration period during day 21 and 30 of life.

**Table 1 metabolites-12-00847-t001:** Evolution of hormonal assays during hospitalization and at 11 months old. PMA: postmenstrual age; GH: growth hormone; IGF1: insulin-like factor 1; D: days; mo: months-old.

Age	Days	D23	D28	D40	D53	D76	D337
**PMA**	Weeks	30 + 2	31 + 0	32 + 5	34 + 4	38 + 0	11 mo
**Glycaemia**	mmol/L mg/dL	1.2 21	4.2 76	3.6 65	3.6 65	4.2 76	4.3 77
**Cortisol**	µg/dL	0.8	1.3	/	/	/	/
**GH**	mUI/L	2.4	5.7	29.9	/	21.3	1.26
**IGF1**	ng/mL	15	21	48	46	82	119
**Insulin**	pmol/L	5903	1625	194	226	82	15.3
**Peptide C**	pmol/L	5.99	3.4	0.9	0.93	0.46	0.36
**Pro-insulin**	pmol/L	159	/	27.7	28	16.4	3.5
**Betamethasone**	mg/kg/d	0.3	0.1	0	0	0	0

## Data Availability

Not applicable.

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
