# Peer review of "Glucocorticoid-Induced Hyperinsulinism in a Preterm Neonate with Inherited ABCC8 Variant"

_metabolites, 2022, doi:10.3390/metabo12090847_

Round 1

Reviewer 1 Report

This is an interesting case report of cortisol--induced hyperinsulinism and hypoglycemia in a pre-term infant. The paper is well writteN and contains original data

Author Response

We thank the reviewer for his nice comments.

We provide in attached file a revised manuscript , with minor modifications, to respond to the few suggestions of another reviewer. 

We hope this will allow a publication of this article.

Best regards,

Dr Emmanuelle MOTTE-SIGNORET, and other co-authors.

Reviewer 2 Report

please see the attached file

Author Response

Comments of the reviewer : 

I think more discussion about the clinical impact and the comparison with the existing studies (especially hyperinsulinemia in premature babies and diabetes mellitus) should be added. In addition, English should be polished before it can be published.
Overall, the finding of the presented case will be attractive for the readers. However, more deeply discussion of the findings is required.

Response to the reviewer:

We thank the reviewer for his nice comments. We provide a revised manuscript according to his suggestions in attached file.

First, this revised version has been corrected by a native-english speaker. Second, we added some new hypothesis in the discussion about the possibility for the hyperinsulinism to have been hidden by classical insulin resistance of EPI for the first three weeks. We also added two references about the programming of metabolic syndrome in low birth weight neonates according the Barker hypothesis.  

We hope that these corrections will allow the publication of this manuscript.

Best regards,

Dr Emmanuelle MOTTE-SIGNORET, and other co-authors

Reviewer 3 Report

The authors reported an interesting case and the data are well analyzed and discussed.

Author Response

(The authors gave the same response as above.)
